# C4-Alkylamination of C4-Halo-1*H*-1-tritylpyrazoles Using Pd(dba)_2_ or CuI

**DOI:** 10.3390/molecules25204634

**Published:** 2020-10-12

**Authors:** Yoshihide Usami, Yuya Tatsui, Hiroki Yoneyama, Shinya Harusawa

**Affiliations:** Department of Pharmaceutical Organic Chemistry, Osaka University of Pharmaceutical Sciences, 4-20-1 Nasahara, Takatsuki, Osaka 569-1094, Japan; e18902@gap.oups.ac.jp (Y.T.); yoneyama@gly.oups.ac.jp (H.Y.); harusawa@gly.oups.ac.jp (S.H.)

**Keywords:** amination, 4-halopyrazole, Buchwald-Hartwig coupling, Pd(dba)_2_, CuI mediated coupling, aliphatic amine

## Abstract

Alkylamino coupling reactions at the C4 positions of 4-halo-1*H*-1-tritylpyrazoles were investigated using palladium or copper catalysts. The Pd(dba)_2_ catalyzed C-N coupling reaction of aryl- or alkylamines, lacking a β-hydrogen atom, proceeded smoothly using *^t^*BuDavePhos as a ligand. As a substrate, 4-Bromo-1-tritylpyrazole was more effective than 4-iodo or chloro-1-tritylpyrazoles. Meanwhile, the CuI mediated C-N coupling reactions of 4-iodo-1*H*-1-tritylpyrazole were effective for alkylamines possessing a β-hydrogen atom.

## 1. Introduction

Synthetic methodologies towards a range of substituted pyrazoles have been developed, as they commonly exhibit bioactivities such as antitumor, antiviral, and antifungal activities. Furthermore, the synthetic study of pyrazoles provides diverse building blocks for the discovery of new drugs, biological probes, herbicides, and other new useful materials [1,2,3]. Therefore, introduction of various functional groups at specific positions on a pyrazole ring is an important and attractive endeavor in synthetic organic chemistry. In particular, the synthesis of C4-aminated pyrazoles has become a prominent research topic, due to the important bioactivities exhibited by this compound class, as shown in Figure 1.

Simple 4-alkylaminopyrazoles (**a** and **b**) have been reported to exhibit weak inhibitory activities against horse lever alcohol dehydrogenase (LADH) [4,5]. Azaisoindolinone derivative (**c**) exhibits potent lipid kinase phosphoinositide 3-kinase γ (PI3Kγ) inhibition, with the distinct advantage of being orally administered and central nervous system (CNS)-penetrant [6]. Two 4-heteroarylamidopyrazoles (**d** and **e**) have been presented as apoptosis signal-regulating kinase 1 (ASK 1) inhibitors [7]. 1-Acetoanilide-4-aminopyrazole-substituted quinazoles are selective Aurora B protein kinase inhibitors with potent anti-tumor activity, and structure **f** is the most potent among them. The 3-aminopyrazole analog of compound **f** is AZD1152, which was the first Aurora B selective inhibitor to enter clinical trials [8]. 7*H*-pyrrolo[2,3*-d*]pyrimidine-based 4-amino-(1*H*)-pyrazole derivative (**g**) and pyrimidine-based 4-amino-(1*H*)-pyrazole derivatives (**h** and **i**) are Janus kinase (JAK) inhibitors. Specifically, compound **i**, a dual inhibitor of JAK and histone deacetylase (HDAC), comprises a zinc-binding moiety (HONHCO) linked to the pyrazole N1, via a (CH_2_)_5_-aliphatic chain [9,10].

The well-known and widely utilized Buchwald-Hartwig coupling reaction is one of the most powerful methods for the amination of aromatic rings. Moreover, the applicability and efficiency of the reaction are continually being improved with the design and development of efficient palladium catalysts, precatalysts, and bulky ligands. Numerous combinations of catalysts and ligands exist that are suitable for specific coupling reactions [11,12,13,14,15,16,17,18].

In spite of such developments, there have been only a few reports of Buchwald-Hartwig coupling at the C4 position of pyrazoles. In 2011, the first example involving the C4 coupling of pyrazoles with aromatic amines was reported by Buchwald, as shown in Scheme 1, Equation (1) [19]. In the following year, the same group described the amidation of five-membered heterocycles with aromatic amides, wherein three examples using 1-benzyl-4-bromopyrazoles and one example using 4-bromo-1-methylpyrazole were reported (Equation (2)) [20]. In their subsequent study on the amination of unprotected five-membered bromoheterocycles, Pd-catalyzed coupling reactions of 4-bromopyrazole with eleven aromatic amines, as well as one benzylic amine, were disclosed (Equation (3)) [21]. Recently, Buchwald et al. described visible-light-mediated amination of aryl halides in the presence of nickel and photoredox catalysts, for which one example of the reaction between 1-benzyl-4-bromopyrazole and pyrrolidine was included (Equation (4)) [22].

In the course of our continuing studies on the functionalization at the C4 position of pyrazoles, we recently reported the synthesis of pyrazole-containing heterobicyclic molecules via ring-closing metathesis [23,24]. Our engagement in pyrazole chemistry has been focused on metal-catalyzed coupling reactions, such as Kumada-Tamao, Suzuki-Miyaura, and Sonogashira couplings, and the Heck-Mizoroki reaction [25,26,27,28]; while the Buchwald-Hartwig coupling reaction for the C4 amination of pyrazoles has remained unchallenged. Encouraged by the above-mentioned successful results, our interest has shifted to Buchwald coupling between 4-halo-1*H*-1-tritylpyrazoles and alkyl amines, which has not been investigated in detail, with readily accessible palladium or copper catalysts, such as bis(benzylideneacetone) palladium(0) (Pd(dba)_2_), or copper (I) iodide (CuI). Herein, we report C4-alkylamino coupling reactions using Pd(dba)_2_ or CuI with 4-halo-1*H*-1-tritylpyrazoles.

## 2. Results and Discussion

### 2.1. Pd(dba)_2_-Catalyzed Buchwald-Hartwig Coupling for C4-Amination of 4-halo-1H-1-tritylpyrazoles

First, we investigated the Buchwald-Hartwig coupling between 4-halo-1*H*-1-tritylpyrazoles (**1**) and piperidine, as a representative secondary amine [16], in order to determine the optimum reaction conditions. The results are summarized in Table 1.

As the Buchwald–Hartwig coupling reaction for 4-halo-1*H*-pyrazoles requires high temperatures (>80 °C) as well as prolonged time [19,20,21], we utilized microwave (MW) apparatus to expedite the experimental process. Ligand screening was performed with the fixed conditions of 4-iodo-1*H*-1-tritylpyrazole (**1_I,_** X = I), Pd(dba)_2_, xylene, 160 °C, and 10 min under MW irradiation (entries 1–4). In the case of commonly used bidentate ligands, namely 1,1’-bis(diphenylphosphino)ferrocene (dppf, **L1**), 1,2-bis(diphenylphosphino)ethane (dppe, **L2**), and 2,2’-bis(diphenylphosphino)diphenyl ether (DPEPhos, **L3**), the reaction did not proceed (entries 1–3), while with the use of the bulky *^t^*BuDavePhos ligand (**L4**) the desired coupled product **2a** was obtained in 21% yield; hence **L4** was deemed a suitable ligand for this coupling reaction (entry 4). The use of **L4** with palladium(II) chloride (PdCl_2_), palladium(II) acetate (Pd(OAc)_2_), or pyridine-enhanced precatalyst preparation stabilization and initiation-isopropyl (PEPPSI-IPr) catalysts did not improve the yield of **2a** (entries 5–7). Although increasing the amount of **L4** to 40 mol% yielded 52% of **2a,** this created an additional problem for the purification of **2a** (entry 8). Solvent screening with the use of **L4** (40 mol%) did not improve results upon that of entry 8 (entries 9–12). Prolonged reaction time (24 h) with **L4** (20 mol%) at room temperature (rt) under MW irradiation gave **2a** in only 7% yield (entry 13). Conducting the reaction at 60 °C and 90 °C afforded **2a** in 19% and 48% yields, respectively (entries 14 and 15). Alternatively, when 4-bromo- and 4-chloropyrazoles **(1_Br_**: X **=** Br and **1_Cl_**: X = Cl) were used as substrates at 90 °C for 24 h (entries 16 and 17), the 4-bromo analogue delivered the highest yield of **2a** (60%) (entry 16). Reaction conditions using bromo compound **1_Br_** at lower or higher temperatures (70 or 140 °C in a sealed reaction vial) delivered inferior results compared to that of entry 16 (entries 18 and 19). Based on these results, further experiments were performed employing the reaction conditions listed in entry 16.

Next, optimized reaction conditions were applied to various amines, and the results are summarized in Table 2. Reactions of **1_Br_** (X=Br) with piperidine and morpholine afforded desired products **2a** and **2b** in 60% and 67% yields, respectively (entries 1 and 2), while the reactions with pyrrolidine and allylamine afforded **2c** (7%) and **2d** (6%) in low yields (entries 3 and 4). The coupling reactions of **1_Br_** with various primary amines produced the corresponding 4-alkylaminopyrazoles **2e–g**, **2k**, and **2l** in low yields (17–34%) (entries 5–8, 11, and 12). Meanwhile, in the cases of isopropylamine and benzylamine, the desired products **2i** and **2j** were not obtained (entries 9 and 10). The reactions of **1_Br_** with adamantylamine or *tert*-butylamine afforded the corresponding products **2m** and **2n** in 90% and 53% yields, respectively (entries 13 and 14). Furthermore, reactions with aromatic amines (anilines and 1-naphtylamine) gave the corresponding **2o** (94%)**, 2p** (91%), and **2q** (85%) in high yields (entries 15–17) as being analogous to Buchwald’s findings [21]. As the reaction with diphenylamine afforded **2r** in 45% yield, we surmised that bulkiness at the reaction center depresses the chemical yield (entry 18). 

Reactions of **1_Br_** with pyrrolidine, allylamine, or primary amines bearing a β-hydrogen atom resulted in low yields (entries 3–12), while amines lacking a β-hydrogen afforded good yields (entries 13–18). These contrasting results are likely due to β-elimination occurring in the palladium complex during the coupling process.

### 2.2. CuI-Catalyzed Coupling for C4-Amination of 4-Halo-1H-1-tritylpyrazoles

Copper-catalyzed C-N coupling reactions have been extensively studied [29], and Buchwald has reportedly implemented this type of reaction using bromo- or iodobenzenes as substrates progressively, but not with five-membered heterocyclic compounds such as pyrazoles [30,31,32,33,34,35,36]. As the C-N coupling reaction of 4-halopyrazoles **1** with allyl- or alkylamines bearing β-hydrogen atoms revealed low reactivities in the above investigation (Table 2, entries 4–12), the copper-catalyzed reaction of **1** was further studied. 

For this purpose, the reaction of allylamine with 4-iodopyrazole **1_I_** (X = I), which could be got easier than 4-bromopyrazole, was investigated, as presented in Table 3. First, the reaction was performed using the conditions similar to those used in Buchwald’s procedure [32]: CuI (5 mol%), 2-isobutyrylcyclohexanone (**L5**: 20 mol%) as the ligand, *N,N*-dimethylformamide (DMF), 100 °C, 24 h, and *t*-BuOK (2 eq). Although the desired 4-allylaminopyrazole **2d** was obtained in only 17% yield (entry 1), increasing the amount of CuI from 5 to 20 mol% improved the chemical yield of **2d** to 72% (entry 2). The use of 2-acetylcyclohexanone (**L6**) as an alternative ligand, which is nearly 10-fold cheaper than **L5,** afforded a good yield (68%, entry 3), while the use of 3,4,7,8-tetramethyl-1,10-phenanthroline (**L7**) resulted in a poor yield (12%, entry 4). Hence, **L6** was applied in the following experiments (entries 5–15 in Table 3). The reaction temperature was varied in entries 5–7, however 100 °C proved optimal (entry 3). Furthermore, various copper catalysts were investigated in entries 9–13, and it was found that the use of the high-cost (CuOTf)_2_·C_6_H_6_ catalyst (entry 13) furnished a comparable yield (70%) to that of CuI (72%) (entry 2). In addition, while the use of 4-bromopyrazole **1** (X **=** Br) provided **2i** in 66% yield (entry 14), chloropyrazole **1_Cl_** (X **=** Cl) did not react (entry 15). 

Therefore, to evaluate the scope of this transformation, additional coupling reactions between iodopyrazole **1_I_** and various amines were performed, by applying the optimized reaction conditions (entry 3 of Table 3), as shown in Table 4. It should be noted that there were a number of distinct contrasts between the outcomes of the CuI-catalyzed (Table 4) and those of the Pd-catalyzed coupling reactions (Table 2). In the case of CuI coupling, reactions of **1i** with piperidine and morpholine afforded **2a** and **2b** (21% and 22%, respectively) in lower yields (Table 4, entries 1 and 2) than those obtained (60% and 67%, respectively) in the corresponding Pd-catalyzed reaction of **1_Br_** (entries 1 and 2 in Table 2). The CuI catalyst provided the pyrrolidine derivative **2c** in 43% yield (Table 4, entry 3), while the Pd catalyst yielded **2c** in only 7% yield (Table 2, entry 3). CuI-catalyzed reactions with primary alkylamines gave moderate to good yields of products **2d**–**2l** (entries 4–12), while reactions with adamantyl, *tert*-butyl, and aromatic amines did not afford the desired products (entries 13–17), and only aniline furnished a low yield of **2o** (15%) (entry 15); these trends were reversed in the case of Pd-catalyzed processes. These negative results may be ascribed to the increase in bulkiness as well as a decrease in the basicity of the amine sources. 

## 3. Conclusions

We have studied the C4 amination of pyrazole derivatives using readily accessible Pd(dba)_2_ or CuI catalysts. The Pd(dba)_2_-catalyzed reaction of 4-bromo-1*H*-1-tritylpyrazole proved to be suitable for aromatic or bulky amines lacking β-hydrogen atoms, but not for cyclic amines (piperidine and morpholine); additionally it was not suitable for alkylamines possessing β-hydrogen atoms. On the other hand, the CuI-catalyzed amination using 4-iodo-1*H*-1-tritylpyrazole was revealed to be favorable for alkylamines possessing β-hydrogen atoms, and not suitable for aromatic amines and bulky amines lacking β-hydrogens, indicating the complementarity of the two catalysts. Although further improvements are required for practical synthesis, such as the reduction of catalyst or ligand loading, the findings of the present study offer a useful synthetic method for the construction of 4-functionalized pyrazoles. Further application of the methodology developed in this study to the C-O coupling reaction of halopyrazoles with alkylated alcohols will be evaluated and reported in the near future.

## 4. Materials and Methods

General: Nuclear magnetic resonance (NMR) spectra were recorded at 27 °C on an Agilent 400-MR-DD2 spectrometer (Agilent Tech., Inc., Santa Clara, CA, USA) in CDCl_3_ with tetramethylsilane (TMS) as an internal standard. Abbreviations for splitting patterns in ^1^H-NMR spectra are noted as d = doublet; t = triplet; q = quartet; quin = quintet; sept = septet. Electron impact-high-resolution mass spectra (EI-HRMS) were measured with a JEOL JMS-700 (2) mass spectrometer (JEOL, Tokyo, Japan). Melting points were determined on a Yanagimoto micromelting point apparatus and were uncorrected. Liquid column chromatography was conducted with silica gel (FL-60D, Fuji Silysia Chemical Ltd., Kasugai, Aichi, Japan). Analytical thin layer chromatography (TLC) was performed on silica gel 70 F_254_ plates (Wako Pure Chemical Industries, Tokyo, Japan), and compounds were detected by dipping the plates into an EtOH solution of phosphomolybdic acid followed by heating. MW-aided reactions were carried out in a Biotage Initiator^®^ reactor (PartnerTech Atvidaberg AB for Biotage Sweden AB, Uppsala, Sweden). Pd(dba)_2_, mesitylene, dppf (**L2**), copper (I) thiophene-2-carboxylate (CuCT), piperidine, pyrrolidine, allylamine, *n*-propylamine, isobutylamine, isoamylamine, isopropylamine, benzylamine, 2-phenylethylamine, 3-phenylpropylamine, adamantylamine, *tert-*butylamine, aniline, 2-methoxyaniline, 1-naphthylamine, and *N,N*-diphenylamine were purchased from Tokyo Chemical Industry (TCI) Co. (Tokyo, Japan). *^t^*BuOK, CuI, and 3,4,7,8-tetramethyl-1,10-phenanthroline (**L7**) were purchased from Nacalai Tesque, Inc. (Kyoto, Japan). Dry xylene, THF, 1,4-dioxane, and DMF were purchased from FUJIFILM Wako Pure Chemical Co. (Osaka, Japan). PEPSI-IPr, dppf (**L1**), DEPPhos (**L3**), *^t^*BuDavePhos (**L4**), morpholine, 2-isobutyrylcyclohexanone (**L5**), and 2-acetylcyclohexanone (**L6**) were purchased from Sigma-Aldrich Co. LLC (St. Louis, MI, USA).

Palladium-catalyzed coupling reaction with 1 and amines (Table 1 and Table 2)

*Typical procedure* (Table 1, entry 16): To a solution of **1_Br_** (50.0 mg, 1.28 × 10^−1^ mmol) in xylene (2 mL) in a MW vial were added *^t^*BuDavePhos (8.8 mg, 2.56 × 10^−2^ mmol, 20 mol%), Pd(dba)_2_ (7.4 mg, 1. 28 × 10^−2^ mmol, 10 mol%), potassium *t*-butoxide (*^t^*BuOK) (28.8 mg, 2.57 × 10^−1^ mmol, 2.0 Equation) and piperidine (0.03 mL, 2.57 × 10^−1^ mmol, 2.0 Equation). The reaction vial was sealed and heated at 90 °C with stirring in an oil bath for 24 h. The reaction mixture was quenched by the addition of sat. aq. NH_4_Cl (1 mL) and extracted with CH_2_Cl_2_ (1 mL × 3). The combined organic layers were dried over MgSO_4_, filtered, and evaporated to give a crude residue, which was purified by silica gel column chromatography (eluent: Hexane/AcOEt = 4:1) to afford 1-(1-trityl-1*H*-pyrazol-4-yl)piperidine (**2a**) (30.9 mg, 60%) as a white powder.

CuI-catalyzed coupling reaction with 1 and amines (Table 3 and Table 4)

*Typical procedure* (Table 3, entry 3); To a solution of **1_I_** (50.0 mg, 1.15 × 10^−1^ mmol) in DMF (2 mL) in a MW vial, were added 2-acetylcyclohexanone (3.0 μL, 2.30 × 10^−2^ mmol, 20 mol%), CuI (4.4 mg, 2.30 × 10^−2^ mmol, 20 mol%), *^t^*BuOK (25.7 mg, 2.30 × 10^−1^ mmol, 2.0 Equation) and allylamine (0.03 mL, 2.30 × 10^−1^ mmol, 2.0 Equation). The reaction vial was sealed and heated at 100 °C with stirring in an oil bath for 24 h. The reaction mixture was quenched by the addition of sat. aq. NH_4_Cl (1 mL) and extracted with CH_2_Cl_2_ (1 mL × 3). The combined organic layers were dried over MgSO_4_, filtered, and evaporated to give a crude residue, which was purified by silica gel column chromatography (eluent: Hexane/AcOEt = 4:1) to afford 2d (28.6 mg, 68%).

*1-(1-Trityl-1H-pyrazol-4-yl)piperidine* (**2a**): white powder; mp 170–174 °C; ^1^H-NMR (400 MHz, CDCl_3_): δ 1.49 (2H, quin, *J* = 5.7 Hz, -CH_2_C*H*_2_CH_2_-), 1.64 (4H, quin, *J* = 5.7 Hz, -CH_2_C*H*_2_CH_2_), 2.83 (4H, t, *J* = 5.7 Hz, -NC*H*_2_CH_2_) 6.88 (1H, d, *J* = 0.8 Hz, pyrazole-H), 7.13–7.18 (6H, m, Ph-H), 7.28–7.31 (9H, m, Ph-H), 7.39 (1H, d, *J* = 0.8 Hz, pyrazole-H); ^13^C-NMR (100 MHz, CDCl_3_): δ 23.9, 25.5, 52.4, 78.4, 118.9, 127.5, 127.6, 129.5, 130.1, 137.7, 143.4; EI-HRMS *m*/*z* calcd. for C_27_H_27_N_3_ (M^+^) 393.2205, found 393.2210.

*4-(1-Trityl-1H-pyrazol-4-yl)morpholine* (**2b**): white powder; mp 209–211 °C; ^1^H-NMR (400 MHz, CDCl_3_): δ 2.87 (4H, t, *J* = 4.7 Hz, -NC*H*_2_CH_2_), 3.78 (4H, t, *J* = 4.7 Hz, -OCH_2_C*H*_2_-), 6.90 (1H, s, pyrazole-H), 7.14–7.17 (6H, m, Ph-H), 7.27–7.30 (9H, m, Ph-H), 7.39 (1H, s, pyrazole-H); ^13^C-NMR (100 MHz, CDCl_3_): δ 51.4, 66.5, 78.5, 118.8, 127.7, 129.0, 130.1, 136.9, 143.3 (two signals are overlapping to give one signal ); EI-HRMS *m*/*z* calcd. for C_25_H_25_N_3_O (M^+^) 395.1996, found 395.1997.

*4-(Pyrrolidin-1-yl)-1-trityl-1H-pyrazole* (**2c**): white powder; mp 189–190 °C; ^1^H-NMR (400 MHz, CDCl_3_): δ 1.90 (4H, br t, *J* = 6.5 Hz, -NCH_2_C*H*_2_-), 2.99 (4H, t, *J* = 6.5 Hz, -NC*H*_2_CH_2_), 6.74 (1H, d, *J* = 0.8 Hz, pyrazole-H), 7.16–7.18 (6H, m, Ph-H), 7.25–7.30 (10H, m, Ph-H and pyrazole-H); ^13^C-NMR (100 MHz, CDCl_3_): δ 24.7, 51.0, 78.3, 116.9, 127.5, 127.6, 128.2, 130.1, 135.1, 143.5; EI-HRMS *m*/*z* calcd. for C_26_H_25_N_3_ (M^+^) 379.2049, found 379.2048.

*N-Allyl-1-trityl-1H-pyrazol-4-amine* (**2d**): oil; ^1^H-NMR (400 MHz, CDCl_3_): δ 3.53 (2H, dt, *J* = 5.7, 1.6 Hz, -NHC*H*_2_CH=CH_2_), 5.09–5.12 (1H, dq, *J =* 10.1, 1.4 Hz, -NHCH_2_CH=CH*H*), 5.16–5.21 (1H, dq, *J =* 17.1, 1.6 Hz, -NHCH_2_CH=C*H*H), 5.86–5.96 (1H, ddt, *J* = 17.1, 10.1, 5.7 Hz, -NHCH_2_C*H*=CH_2_), 6.88 (1H, d, *J* = 0.8 Hz, pyrazole-H), 7.14–7.18 (6H, m, Ph-H), 7.25–7.30 (9H, m, Ph-H), 7.32 (1H, d, *J =* 0.8, pyrazole-H); ^13^C-NMR (100 MHz, CDCl_3_): δ 50.5, 78.3, 116.3, 119.2, 127.6, 129.9, 130.1, 132.3, 135.8, 143. 4; EI-HRMS *m*/*z* calcd. for C_25_H_23_N_3_ (M^+^) 365.1892, found 365.1892.

*N-Propyl-1-trityl-1H-pyrazol-4-amine* (**2e**): white powder; mp 145–148 °C; ^1^H-NMR (400 MHz, CDCl_3_): δ 0.94 (3H, t, *J* = 7.4 Hz, -NHCH_2_CH_2_C*H*_3_), 1.56 (2H, sext, *J* = 7.4 Hz, -NHCH_2_C*H*_2_CH_3_), 2.86 (2H, t, *J* = 7.0 Hz, -NHC*H*_2_CH_2_CH_3_), 6.86 (1H, d, *J* = 0.8 Hz, pyrazole-H), 7.14–7.20 (6H, m, Ph-H), 7.26–7.35 (10H, m, Ph-H and pyrazole-H); ^13^C-NMR (100 MHz, CDCl_3_): δ 11.6, 23.0, 49.7, 78.2, 118.7, 127.5, 127.6, 129.7, 130.1, 132.9, 143.5; EI-HRMS *m*/*z* calcd. for C_25_H_25_N_3_ (M^+^) 367.2048, found 367.2049.

*N-Butyl-1-trityl-1H-pyrazol-4-amine* (**2f**): white amorhous; mp 112–116 °C; ^1^H-NMR (400 MHz, CDCl_3_): δ 0.91 (3H, t, *J* = 7.4 Hz, -CH_2_C*H*_3_-), 1.36 (2H, br sext, *J* = 7.4 Hz, -CH_2_C*H*_2_CH_3_), 1.52 (2H, br quint, *J* = 7.4 Hz, -CH_2_C*H*_2_CH_2_-),2.89 (2H, t, *J* = 7.0 Hz, -NHC*H*_2_CH_2_-), 6.86 (1H, s, pyrazole-H), 7.14–7.19 (6H, m, Ph-H), 7.27–7.33 (10H, m, Ph-H and pyrazole-H); ^13^C-NMR (100 MHz, CDCl_3_): δ 14.0, 20.2, 32.0, 47.6, 78.2, 118.7, 127.5, 127.6, 129.7, 130.1, 132.9, 143.4 EI-HRMS *m*/*z* calcd. for C_26_H_27_N_3_ (M^+^) 381.2205, found 381.2215.

*N-Isobutyl-1-trityl-1H-pyrazol-4-amine* (**2g**): white powder; mp 135–136 °C; ^1^H-NMR (400 MHz, CDCl_3_): δ 0.93 (6H, d, *J* = 6.6 Hz, -NHCH_2_CH(C*H*_3_)_2_), 1.78 (1H, nonet, *J* = 6.6 Hz, -NHCH_2_C*H*(CH_3_)_2_), 2.70 (2H, d, *J* = 6.6 Hz, -NHC*H*_2_CH(CH_3_)_2_), 6.85 (1H, s, pyrazole-H), 7.11–7.19 (6H, m, Ph-H), 7.25–7.32 (10H, m, Ph-H and pyrazole-H); ^13^C-NMR (100 MHz, CDCl_3_): δ 20.5, 28.4, 55.7, 78.2, 118.4, 127.5, 127.6, 129.6, 130.1, 133.1, 143.5; EI-HRMS *m*/*z* calcd. for C_26_H_27_N_3_ (M^+^) 381.2205, found 381.2210.

*N-Isoamyl-1-trityl-1H-pyrazol-4-amine* (**2h**): white amorphous; mp 110–113 °C; ^1^H-NMR (400 MHz, CDCl_3_): δ 0.89 (6H, d, *J* = 6.7 Hz, -CH(C*H*_3_)_2_), 1.48 (2H, q, *J* = 7.4 Hz, -CH_2_C*H*_2_CH-), 1.64 (1H, nonet, *J* = 6.6 Hz, -CH_2_C*H*(CH_3_)_2_), 2.89 (2H, br t, *J* = 7.3 Hz, -NHC*H*_2_CH_2_-), 6.86 (1H, s, pyrazole-H), 7.15–7.18 (6H, m, Ph-H), 7.26–7.32 (10H, m, Ph-H and pyrazole-H); ^13^C-NMR (100 MHz, CDCl_3_): δ 22.6, 25.9, 38.9, 46.0, 78.3, 118.7, 127.5, 127.6, 129.7, 130.1, 132.9, 143.5; EI-HRMS *m*/*z* calcd. for C_27_H_29_N_3_ (M^+^) 395.2361, found 395.2359.

*N-Isopropyl-1-trityl-1H-pyrazol-4-amine* (**2i**): white powder; mp 130–133 °C; ^1^H-NMR (400 MHz, CDCl_3_): δ 1.11 (6H, d, *J* = 6.3 Hz, -NHCH(C*H*_3_)_2_), 3.18 (1H, sept, *J* = 6.3 Hz, -NHC*H*(CH_3_)_2_), 6.88 (1H, s, pyrazole-H), 7.15–7.19 (6H, m, Ph-H), 7.26–7.35 (10H, m, Ph-H and pyrazole-H); ^13^C-NMR (100 MHz, CDCl_3_): δ 23.0, 48.4, 78.2, 120.5, 127.5, 127.6, 130.1, 131.1, 143.4 (two carbon signals overlapped); EI-HRMS *m*/*z* calcd. for C_25_H_25_N_3_ (M^+^) 367.2048, found 367.2046

*N-Benzyl-1-trityl-1H-pyrazol-4-amine* (**2j**): white powder; mp 148–151 °C; ^1^H-NMR (400 MHz, CDCl_3_): δ 4.06 (2H, s, -C*H*_2_Ph), 6.84 (1H, s, pyrazole-H), 7.13–7.16 (6H, m, Ph-H), 7.24–7.30 (14H, m, Ph-H), 7.32 (1H, s, pyrazole-H); ^13^C-NMR (100 MHz, CDCl_3_): δ52.2, 78.3, 119.2, 127.2, 127.5, 127.6, 127.9, 128.5, 129.9, 130.1, 132.4, 139.4, 143.4; EI-HRMS *m*/*z* calcd. for C_29_H_25_N_3_ (M^+^) 415.2048, found 415.2046.

*N-Phenethyl-1-trityl-1H-pyrazol-4-amine* (**2k**): white powder; mp 134–137 °C; ^1^H-NMR (400 MHz, CDCl_3_): δ 2.84 (2H, t, *J* = 6.9 Hz, -NHCH_2_C*H*_2_Ph), 3.16 (2H, t, *J* = 6.9 Hz, -NHC*H*_2_CH_2_Ph), 6.85 (1H, d, *J* = 0.9 Hz, pyrazole-H), 7.14–7.32 (21H, m, Ph-H and pyrazole-H); ^13^C-NMR (100 MHz, CDCl_3_) δ 35.8, 48.9, 78.3, 119.0, 126.4, 127.5, 127.6, 128.6, 128.8, 129.8, 130.1, 132.3, 139.3, 143.4; EI-HRMS *m*/*z* calcd. for C_30_H_27_N_3_ (M^+^) 429.2205, found 429.2200.

*N-(3-Phenyl)propyl-1-trityl-1H-pyrazol-4-amine* (**2l**): white powder; mp 114–117 °C; ^1^H-NMR (400 MHz, CDCl_3_): δ 1.86 (2H, br quint, *J* = 7.3 Hz, -CH_2_C*H*_2_ CH_2-_), 2.67 (2H, br t, *J* = 7.5 Hz, -CH_2_C*H*_2_Ph), 2.94 (2H, t, *J* = 7.1 Hz, -NHC*H*_2_CH_2_-), 6.84 (1H, s, pyrazole-H), 7.14–19 (8H, m, Ph-H and pyrazole-H), 7.24–7.30 (13H, m, Ph-H, and pyrazole-H); ^13^C-NMR (100 MHz, CDCl_3_) δ 31.4, 33.3, 47.4, 78.3, 118.8, 125.9, 127.5, 127.6, 128.3, 128.4, 129.8, 130.1, 132.6, 141.8, 143.4; EI-HRMS *m*/*z* calcd. for C_31_H_29_N_3_ (M^+^) 443.2362, found 443.2365.

*N-((3s,5s,7s)-Adamantan-1-yl)-1-trityl-1H-pyrazol-4-amine* (**2m**): white powder; mp 204–205 °C; ^1^H-NMR (400 MHz, CDCl_3_): δ 1.59 (12H, m, Ad-H), 2.05 (4H, br n, Ad-H, and -N*H*Ad), 7.00 (1H, s, pyrazole), 7.14–7.18 (6H, m, Ph-H), 7.28–7.30 (9H, m, Ph-H), 7.34 (1H, s, pyrazole-H); ^13^C-NMR (100 MHz, CDCl_3_): δ 29.6, 36.4, 43.2, 51.7, 78.3, 125.4, 127.2, 127.5, 127.6, 130.1, 136.8, 143.3; EI-HRMS *m*/*z* calcd. for C_32_H_33_N_3_ (M^+^) 459.2674, found 459.2673.

*N-(tert-Butyl)-1-trityl-1H-pyrazol-4-amine* (**2n**): white powder; mp 137–140 °C; ^1^H-NMR (400 MHz, CDCl_3_): δ 1.11 (9H, s, -C(C*H*_3_)_3_), 7.01 (1H, s, pyrazole-H), 7.15–7.18 (6H, m, Ph-H), 7.28–7.30 (9H, m, Ph-H), 7.36 (1H, s, pyrazole-H); ^13^C-NMR (100 MHz, CDCl_3_): δ 29.5, 51.9, 78.3, 126.7, 127.0, 127.5, 127.6, 130.1, 136.2, 143.3; EI-HRMS *m*/*z* calcd. for C_26_H_27_N_3_ (M^+^) 381.2205, found 381.2206.

*N-Phenyl-1-trityl-1H-pyrazol-4-amine* (**2o**): white powder; mp 191–192 °C; ^1^H-NMR (400 MHz, CDCl_3_): δ 5.05 (1H, br, -N*H*Ph), 6.70–6.76 (3H, m, Ph-H and pyrazole-H), 7.14–7.20 (7H, m, Ph-H), 7.24–7.32 (11H, m,Ph-H), 7.61 (1H, s, pyrazole-H); ^13^C-NMR (100 MHz, CDCl_3_): δ 78.8, 113.4, 118.5, 123.5, 127.2, 127.8, 129.3, 130.0, 130.1, 136.0, 143.1, 146.6; EI-HRMS *m*/*z* calcd. for C_28_H_23_N_3_ (M^+^) 401.1892, found 401.1890.

*N-(o-Methoxy)phenyl-1-trityl-1H-pyrazol-4-amine* (**2p**): white powder; mp 133–136 °C; ^1^H-NMR (400 MHz, CDCl_3_): δ 3.87 (3H, s, -OCH_3_), 5.70 (1H, br, -N*H*Ar), 6.70–6.76 (1H, m, Ph-H), 6.82–6.84 (2H, m, Ph-H), 7.22–7.25 (8H, m, Ph-H), 7.32–7.68 (9H, m, Ph-H, pyrazole-H), 7.68 (1H, s, pyrazole-H); ^13^C-NMR (100 MHz, CDCl_3_): δ 55.4, 78.6, 109.8, 110.9, 117.6, 121.1, 123.3, 126.6, 127.6, 130.1, 135.7, 136.2, 143.1, 146.5; EI-HRMS *m*/*z* calcd. for C_29_H_25_N_3_ (M^+^) 431.1998, found 431.1998.

*N-(Naphthalen-1-yl)-1-trityl-1H-pyrazol-4-amine* (**2q**): white powder; mp 175–178 °C; ^1^H-NMR (400 MHz, CDCl_3_): δ 5.66 (1H, s, *-*N*H-*naphthyl), 6.82–6.84 (1H, m, naphtyl-H), 7.21–7.26 (8H, m, Ph-H and naphthyl-H), 7.28–7.36 (9H, m, Ph-H), 7.39 (1H, s, pyrazole-H), 7.43–7.48 (2H, m, naphthyl-H), 7.69 (1H, s, pyrazole-H), 7.79–7.87 (2H, m, naphthyl-H); ^13^C-NMR (100 MHz, CDCl_3_): δ 78.8, 106.9, 118.9, 119.8, 123.4, 123.6, 125.1, 125.9, 126.3, 127.6, 127.8, 128.7, 130.1, 130.4, 134.4, 136.4, 142.2, 143.1; EI-HRMS *m*/*z* calcd. for C_32_H_25_N_3_ (M^+^) 451.2049, found 451.2052.

*N,N-Diphenyl-1-trityl-1H-pyrazol-4-amine* (**2r**)**:** white powder; mp 175–177 °C; ^1^H-NMR (400 MHz, CDCl_3_): δ6.92 (2H, t, *J* = 7.3 Hz, Ph-H), 7.04–7.06 (4H, m, Ph-H and pyrazole-H), 7.16–7.22 (10H, m, Ph-H), 7.29–7.33 (10H, m, Ph-H and pyrazole-H), 7.52 (1H, s, pyrazole-H); ^13^C-NMR (100 MHz, CDCl_3_): δ 78.9, 121.5, 121.9, 127.7, 127.9, 127.74, 127.78, 127.8, 128.6, 129.1, 129.2, 130.1, 130.2, 137.1, 143.0, 147.7; EI-HRMS *m*/*z* calcd. for C_34_H_27_N_3_ (M^+^) 477.2205, found 477.2197.

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
