# Peer review of "C4-Alkylamination of C4-Halo-1H-1-tritylpyrazoles Using Pd(dba)2 or CuI"

_molecules, 2020, doi:10.3390/molecules25204634_

Round 1
Reviewer 1 Report
The manuscript submitted by Usami et al reports their systematic works on the synthesis of 4-aminopyrazoles via the Pd- and Cu-catalyzed amination reactions of 4-halopyrazoles, leading to the efficient synthesis of various 4-aminopyrazoles with sound substrate scope. The work provides complement to those known method on the synthesis of amino-functionalized heteroaryls. The work is recommended for publication in Molecules after minor revision.
1. The authors have shown the related advances in the amination/amidation reactions of 4-bromopyraozles, they should also emphasize and summarize the advantage(s) of their work by comparing the literature methods.
2. General reaction conditions, i.e. the volume of solvent, substrates’ loading etc in the optimization section should be provided as table footnotes.
3. In both the reaction with Pd- and Cu-catalysis, the reactions using ammonium salt such as AcONH4 as NH3 surrogate should be conducted to see if the methods can be used for the synthesis free amino-functionalized pyrazoles.
4. In the experimental section, the “HREIMS” should be more scientifically presented as “EI-HRMS”.
5. A few recent works on copper-catalyzed amination mediated by C-halogen bond should be cited in the section of copper-catalysis: Tetrahedron Lett. 2020, 61, 151683; Org. Chem. Front. 2020, 7, 1107 etc.
Author Response
Thank you for useful suggestions.
Suggestion 1. The authors have shown the related advances in the amination/amidation reactions of 4-bromopyraozles, they should also emphasize and summarize the advantage(s) of their work by comparing the literature methods.
Response: Thank you for suggestion. But we simply wrote that “with readily accessible palladium or copper catalysts, such as bis(benzylideneacetone) palladium(0) [Pd(dba)2], or cupper (I) iodide (CuI)” in last paragraph in introduction. In reported coupling reactions, relatively expensive catalysts and ligands were applied in specific combination to present high chemical yield with require severe anhydrous condition by several operations.
Suggestion 2. General reaction conditions, i.e. the volume of solvent, substrates’ loading etc in the optimization section should be provided as table footnotes.
Response: we added information in table footnotes.
Suggestion 3. In both the reaction with Pd- and Cu-catalysis, the reactions using ammonium salt such as AcONH4 as NH3 surrogate should be conducted to see if the methods can be used for the synthesis free amino-functionalized pyrazoles.
Response: Thank you for useful comments. We will examine the pointsuggested by the reviewer in our future work.
Suggestion 4. In the experimental section, the “HREIMS” should be more scientifically presented as “EI-HRMS”.
Response: we revised as required.
Suggestion 5. A few recent works on bond should be cited in the section of copper-catalysis: Tetrahedron Lett. 2020, 61, 151683; Org. Chem. Front. 2020, 7, 1107 etc.
Response: some references concerning copper-catalyzed amination mediated by C-halogen to were added.
Reviewer 2 Report
molecules-944630
C4-Alkylamination of C4-Halo-1H-1-tritylpyrazoles Using Pd(dba)2 or CuI
Yoshihide Usami *, Yuya Tatsui, Hiroki Yoneyama, Shinya Harusawa
This manuscript deals with the C4-amination of pyrazole derivatives using two different catalytic species, Pd(dba)2 or CuI. The two catalysts showed completely different reactivity in the reactions performed with several amines.
Even if the results are quite interesting, I suggest that the manuscript should be reconsidered after the following revisions:
1) The reactions between CuI and different amines (Table 4) are not discussed. In particular there is not any comment on the results described in entries 13-17.
2) The complete different behaviour of Pd(dba)2 and CuI in the reactions of adamantylamine, tBu-amine, aniline, 1-naphtylamine and N,N-diphenylamine could not be ascribed to their basicity or bulkiness since the same amines have been tested with the two catalyst, but must be connected to the different experimental conditions such as the ligand and the solvent. A few more experiments should be performed in order to clarify this point.
3) dba and PEPPSI-IPr have not been tested so they can be removed from Table 1.
Author Response
Thank you for useful suggestions.
Suggestion 1) The reactions between CuI and different amines (Table 4) are not discussed. In particular, there is not any comment on the results described in entries 13-17.
Response: Thank you for suggestion. Yes, we only commented as “while reactions with adamantyl, tert-butyl, and aromatic amines did not afford the desired products (entries 13-17), and only aniline furnished a low yield of 2o (15%) (entry 15)” and “These negative results may be ascribed to the increase in bulkiness as well as a decrease in the basicity of the amine sources” Comments we can make now are presented.
Suggestion 2) The complete different behaviour of Pd(dba)2 and CuI in the reactions of adamantylamine, tBu-amine, aniline, 1-naphtylamine and N,N-diphenylamine could not be ascribed to their basicity or bulkiness since the same amines have been tested with the two catalyst, but must be connected to the different experimental conditions such as the ligand and the solvent. A few more experiments should be performed in order to clarify this point.
Response: Thank you for suggestion. Whereas more detailed experiments are required to clarify it on each interested amines, we think we presented enough information for primitive reactivity. And our interest is already moving to the following C-O coupling as described at the last sentence in conclusion.
Suggestion 3) dba and PEPPSI-IPr have not been tested so they can be removed from Table 1.
Response: figure for Pd(dba)2 was included instead of dba in footnote of Table 1. PEPPSI-IPr was tested in entry 7.
Reviewer 3 Report
In this paper, the authors reported Pd(dba)2 or CuI catalyzed C4-Alkylamination of 4-Halo-1H-1-tritylpyrazoles. The manuscript is well written, the experiments are straightforward, and the results are well presented.
Other comments:
For microwave-assisted reactions, have the authors attempted a relatively longer reaction time?
Please confirm the structure of L5. As shown in the Materials and Methods, 2-butyrylcyclohexanone was ordered, instead of 2-isobutyrylcyclohexanone.
The abbreviations for splitting patterns and information regarding the internal standard should be included.
Is compound 2m actually compound 2n? If so, please renumber compounds 2m and 2n.
The authors should always use the correct compound names or chemical formulas, for example, KtOBu on page 8.
Please also ensure the accuracy of the data. For example, compound 2h, 1H_NMR, should 1.48 (1H, q, … ) be 1.48 (2H, q, …) ?
Author Response
Thank you for useful suggestions.
Suggestion: For microwave-assisted reactions, have the authors attempted a relatively longer reaction time?
Response: No. Whereas we aimed to shorten reaction time by using microwave heating at higher temperature, but chemical yield of desired coupled compound was not satisfactory. We choose lower temperature and long-time reaction condition reluctantly. Indeed, reaction at 140 oC for 24 h seen in entry 19 in Table 1 gave lower yield.
Suggestion: Please confirm the structure of L5. As shown in the Materials and Methods, 2-butyrylcyclohexanone was ordered, instead of 2-isobutyrylcyclohexanone.
Response: We revised to 2-isobutyrylcyclohexanone
Suggestion: The abbreviations for splitting patterns and information regarding the internal standard should be included.
Response: required information was added.
Suggestion: Is compound 2m actually compound 2n? If so, please renumber compounds 2m and 2n.
Response: 2m and 2n were corrected in Experimental section.
Suggestion: The authors should always use the correct compound names or chemical formulas, for example, KtOBu on page 8.
Response: we made correction.
Suggestion: Please also ensure the accuracy of the data. For example, compound 2h, 1H_NMR, should 1.48 (1H, q, … ) be 1.48 (2H, q, …) ?
Response: it was corrected as 1.48 (2H, q, …).
Round 2
Reviewer 2 Report
The revised manuscript is now ready for publication
Author Response
Thank you very much for your comments.